

# The relationship between disease activity and quality of life in rheumatoid arthritis patients: a network analysis

Ruo-Wei Ma, Lin Zhu, Ming-Hui Zhang, Cui-E Li, Yu-Xuan Zhang, Bin-Bin Feng and Guo-Cui Wu

School of Nursing, Anhui Medical University, Hefei, China

## ABSTRACT

**Background**. The relationship between disease activity and quality of life (QoL) in rheumatoid arthritis (RA) patients was explored using network analysis. The focus of network analysis has recently shifted from studying individual groups to comparing the network structures of different subgroups. RA patients with depressive symptoms generally have lower QoL scores, so we compared the QoL networks of RA patients with and without depressive symptoms to test for differences.

**Methods**. QoL, depressive symptoms, and disease activity were measured using the 36-item Short-Form Health Survey (SF-36), Hospital Anxiety and Depression Scale (HADS), and Disease Activity Score 28 joints (DAS28). A flowchart was drawn to explore the relationship between disease activity and QoL. The RA patients were divided into groups with and without depressive symptoms for network comparison.

**Results**. A total of 424 patients with RA were included in this study. Disease activity was strongly associated with the PF (physical function) domain (edge weight = 0.237). The results of the network comparisons showed differences between the QoL network structures (M = 0.274, $P = 0.029$) and some specific edge strengths.

**Conclusion**. Disease activity is strongly associated with QoL in RA patients, especially in the domain of PF, and the structure of the QoL network changes in the presence of depressive symptoms in RA patients.

Corresponding author
Guo-Cui Wu, gcwu82@163.com

## INTRODUCTION

Rheumatoid arthritis (RA) is one of the most common chronic inflammatory diseases, which can cause joint damage and even disability (*Smolen, Aletaha & McInnes, 2016*). In recent years, the global prevalence of RA has been continuously increasing. In China, the prevalence of RA is approximately 0.28% (*Finckh et al., 2022*). As RA cannot be completely cured, "disease improvement" has become an important prognostic indicator, and quality of life (QoL) is one of the most important indicators (*Smolen et al., 2020*).

In the pathology of RA, cytokines such as IL-1, TNF-alpha, and IL-6 are secreted, which can enter the brain and disrupt the metabolism of monoamine neurotransmitters, affecting normal neurological and cognitive functions and promoting depressive symptoms

(*Liu, Ho, & Mak, 2012*; *Pan et al., 2011*). RA and depressive symptoms have been found to interact in a large longitudinal study, and RA patients with depressive symptoms have lower QoL scores (*Lu et al., 2016*; *Matcham et al., 2014*; *Tański, Szalonka, & Tomasiewicz, 2022*). Network analysis has shifted from studying individual groups to comparing the network structure of different subgroups (*Van Borkulo et al., 2015*). In a study of obesity patients, differences were found between the QoL network structures of patients with low and high physical performance (*Dalle Grave et al., 2020*). However, no study has explored whether there are differences in the network structure of QoL between RA patients with and without depressive symptoms.

According to EULAR guidelines, "remission" is the primary treatment target for RA patients. When "remission" is not feasible, "low disease activity" is accepted to inhibit disease progression (*Smolen et al., 2023*). However, the remission rate for RA patients is relatively low, approximately 5%–45% (*Ajeganova & Huizinga, 2017*). A study in Ireland indicates that RA patients who achieve clinical remission or low disease activity have significantly higher QoL scores than those with moderate or high disease activity (*Murphy et al., 2024*). The same conclusion has been validated in the Portuguese population (*Rosa-Gonçalves, Bernardes & Costa, 2018*). These studies assess the QoL as a single, unified structure. In fact, the QoL is made up of several domains (*Matcham et al., 2014*). The complex relationship between disease activity and these domains is still unclear.

In addition, it may be difficult to accurately assess the relationship between disease activity and QoL because of the strong correlation between QoL domains (*Koh et al., 2006*). Most previous studies have considered QoL as a single overall construct and have not considered correlations between QoL domains (*Bai et al., 2023*; *Zhang et al., 2023*). This makes it difficult to assess the impact of disease activity on different domains of QoL in RA patients based on previous research. Network analysis can control the correlation impact between QoL domains and identify the related domains to disease activity (*Bai et al., 2023*; *Takahashi et al., 2023*).

This study aims to (1) compare whether there is a difference in the structure of the QoL network between RA patients with and without depressive symptoms and (2) explore the relationship between disease activity and QoL in RA patients.

## MATERIAL AND METHODS

### Participants

This study consecutively included patients with RA who attended the Department of Rheumatology and Immunology of the First Affiliated Hospital of Anhui Medical University and met the diagnostic criteria between January 2024 and July 2024. When patients presented to the hospital with RA-related problems, they were asked if they were willing to participate in the study, and 424 patients were finally included. The inclusion criteria were as follows: (1) patients diagnosed with RA by experienced rheumatologists who also fulfilled the 2010 American College of Rheumatology (ACR)/European League Against Rheumatism (EULAR) classification criteria for RA; (2) informed consent and voluntary participation in the study; (3) age of 18 years or older; and (4) the ability to communicate
in Mandarin and complete the questionnaire independently or with assistance from the investigator. The exclusion criteria were the following: (1) comorbidity with other autoimmune diseases, such as systemic lupus erythematosus, Sjögren's syndrome, *etc.*; (2) suffering from systemic diseases that may affect sleep quality, such as fibromyalgia and chronic fatigue syndrome.

## Ethics approval

This study followed the ethical principles of the World Medical Association Declaration of Helsinki on medical research involving human subjects. The Ethics Review Committee of Anhui Medical University approved this study with the IRB number 83243410. All subjects provided written informed consent.

## Measurements

### Demographic and disease characteristics

Demographic and disease characteristics included age, gender, body mass index (BMI), family location, marital status, education level, smoking, alcohol consumption, sleep duration, disease duration, number of chronic diseases, Joint Visual Analog Scale (VAS) score, swollen joint count, tender joint count, and erythrocyte sedimentation rate (ESR).

### Hospital anxiety and depression scale (HADS)

Depressive symptoms in RA patients were measured using the Chinese version of the HADS scale (*Yang et al., 2014*). The scale includes two subscales measuring anxiety and depressive symptoms, and we used the subscale measuring depressive symptoms. Each item has a score range of 0–3 and a total score range of 0–21 (*Traki et al., 2014*). A score of 0–7 indicates no depressive symptoms; 8–10 indicates mild; 11–14 indicates moderate; and 15–21 indicates severe depressive symptoms (*Beekman & Verhagen, 2018*). We divided patients with scores less than or equal to 7 into a group without depressive symptoms and those with scores greater than 7 into a group with depressive symptoms. The Cronbach's coefficient for this scale was 0.85 (*Yang et al., 2014*).

### The 36-item short-form health survey (SF-36)

Assessing the QoL in RA patients using the Chinese version of the SF-36 scale (*Wu et al., 2023*), this scale comprises eight domains: physical function (PF), role-physical (RP), bodily pain (BP), general health (GH), vitality (VT), social function (SF), role-emotional (RE), and mental health (MH). Each domain translates to a score of 0–100, with a higher score indicating a better QoL (*Lin et al., 2020*). The Cronbach's coefficient for this scale was 0.87, indicating good reliability (*Wu et al., 2023*).

### Disease activity score 28 joints (DAS28)

Disease activity was assessed by a rheumatologist using the Disease Activity Score 28 joints (DAS28). The DAS28 score was calculated as follows: DAS28 = $0.56 \times$ sqrt (tender joint count) + $0.28 \times$ sqrt (swollen joint count) + $0.70 \times$ ln (ESR) + $0.014 \times$ PGA (patient global assessment) (*Van der Heijde & Jacobs, 1998*), where ESR reflects the level of inflammatory markers. PGA is a subjective evaluation of the patient's overall condition by means of a

visual analog scale (VAS, 0–100 mm). Higher scores indicate more severe disease activity in RA.

## Statistical analysis

Descriptive analyses of the variables were performed using IBM SPSS Statistics Version 26.0 (IBM Corp., Armonk, NY, USA). Continuous variables were presented as mean ± standard deviation (M ± SD) after the normality test confirmed that they were normally distributed. Continuous variables that were not normally distributed were presented as median (interquartile range, IQR) (median (IQR)). Categorical variables were described as frequencies and percentages (n, %). Network analyses were performed using RStudio (*RStudio Team, 2020*; *Epskamp, Borsboom & Fried, 2018*). The "qgraph" package is used to visualize the network (*Yang et al., 2022*). In the network, variables are considered 'nodes', and 'edges' are formed between 'nodes' using regularised partial correlation methodology. This methodology effectively estimates weak or unstable correlations as zero, resulting in a sparse graph that only represents robust relationships (*Dalle Grave et al., 2020*). The Extended Bayesian Information Criterion (EBIC) is used to select the best-fitting model (*Zhang, Zhang & Wells, 2008*).

The RA patients were divided into groups with and without depressive symptoms. Expected influence (EI) values were used to identify the most influential domain of the QoL network (*Song et al., 2024*). The "NetworkComparisonTest" (NCT) package was used to detect any differences between the two networks (*Forbes et al., 2021*). We also created a flowchart using the "flow" function in the "qgraph" package to explore the relationship between disease activity and QoL, and the predictability of each node using the "mgm" package (*Zhang, Zhu & Wu, 2023*).

Two analyses were performed utilizing the "bootnet" package to evaluate the stability and accuracy of the network. The first was an estimation of EI stability based on a case-drop bootstrap procedure (1,000 iterations), while the second was an assessment of the 95% confidence intervals (CIs) of edge weights based on a nonparametric bootstrap procedure (1,000 iterations) (*Epskamp, Borsboom & Fried, 2018*). A correlation stability coefficient (CS-C) of EI greater than 0.50 indicates that the stability of the network is relatively ideal; narrower 95% CIs signify a more precise network (*Mullarkey, Marchetti & Beevers, 2019*).

## RESULTS

A total of 424 RA patients were included in this study. Table 1 shows the demographic and disease characteristics of RA patients without ($n = 235$) and with ($n = 189$) depressive symptoms. The participants had a mean age of $60.22 \pm 12.15$ years (mean ± SD). Most were female (77.6%). The two groups differed in age, family location, education level, number of chronic diseases, sleep duration, joint VAS scores, swollen joint count, tender joint count, and ESR.

Figure 1A shows the QoL network of patients without depressive symptoms, and Fig. 1B shows the network structure of patients with depressive symptoms. Both network diagrams have 28 edges, of which 25 are non-zero edges, reflecting the high density of the network.

**Table 1  Participant characteristics ($n = 424$).**

| Variables | M ± SD or n (%) | Without depressive symptoms ($n = 235$) M ± SD or n (%) | With depressive symptoms ($n = 189$) M ± SD or n (%) | F/t/χ² | P |
|---|---|---|---|---|---|
| Age (years) | 60.22 ± 12.15 | 58.21 ± 12.41 | 62.74 ± 11.10 | 14.96 | **<0.001** |
| Gender | | | | 1.01 | 0.308 |
| Male | 95 (22.4) | 57 (24.3) | 38 (20.1) | | |
| Female | 329 (77.6) | 178 (75.7) | 151 (79.9) | | |
| BMI (kg/m²) | 21.88 ± 3.99 | 22.07 ± 3.76 | 21.65 ± 4.24 | 1.15 | 0.284 |
| Family location | | | | 6.01 | **0.014** |
| Rural | 269 (63.4) | 137 (58.3) | 132 (69.8) | | |
| Urban | 155 (36.6) | 98 (41.7) | 57 (30.2) | | |
| Marital status | | | | 1.46 | 0.690 |
| Unmarried | 5 (1.2) | 3 (1.3) | 2 (1.1) | | |
| Married | 394 (92.9) | 221 (94.0) | 173 (91.5) | | |
| Divorced | 2 (0.5) | 1 (0.4) | 1 (0.5) | | |
| Widowed | 23 (5.4) | 10 (4.3) | 13 (6.9) | | |
| Education level | | | | 27.82 | **<0.001** |
| Illiterate | 102 (24.1) | 37 (15.7) | 65 (34.4) | | |
| Elementary school | 168 (39.6) | 92 (39.1) | 76 (40.2) | | |
| Junior high school | 106 (25.0) | 70 (29.8) | 36 (19.2) | | |
| High school | 41 (9.7) | 30 (12.8) | 11 (5.8) | | |
| University | 7 (1.7) | 6 (2.6) | 1 (0.4) | | |
| Smoking | | | | 0.08 | 0.768 |
| Yes | 87 (20.5) | 47 (20.0) | 40 (21.2) | | |
| No | 337 (79.5) | 188 (80.0) | 149 (78.8) | | |
| Alcohol consumption | | | | 0.21 | 0.648 |
| Yes | 57 (13.4) | 30 (12.8) | 27 (14.3) | | |
| No | 367 (86.6) | 205 (87.2) | 162 (85.7) | | |
| Disease duration (years) | 10.96 ± 10.01 | 10.13 ± 9.39 | 11.99 ± 10.66 | 3.63 | 0.057 |
| Number of chronic diseases | 0.96 ± 0.91 | 0.88 ± 0.93 | 1.08 ± 0.86 | 5.33 | **0.021** |
| Sleep duration (hours) | 6.17 ± 1.81 | 6.58 ± 1.71 | 5.67 ± 1.81 | 28.39 | **<0.001** |
| Joint VAS score (mm) | 47.25 ± 27.09 | 38.87 ± 23.73 | 57.68 ± 27.45 | 57.14 | **<0.001** |
| Swollen joint count | 3.71 ± 4.83 | 2.99 ± 4.21 | 4.61 ± 5.39 | 12.02 | **<0.001** |
| Tender joint count | 6.92 ± 7.22 | 5.23 ± 6.10 | 9.02 ± 7.94 | 30.84 | **<0.001** |
| ESR (mm/h) | 58.16 ± 38.23 | 53.85 ± 37.05 | 63.51 ± 39.10 | 6.77 | **0.010** |

**Notes.**

Categorical variables were expressed using the number of cases (%), and continuous variables were described using the mean ± standard deviation (M ± SD).

BMI, Body Mass Index; Joint VAS score, Joint Visual Analog Scale (VAS) score.

"Alcohol consumption yes" was defined based on the Chinese Dietary Guidelines for Residents as follows: For male adults, weekly pure alcohol intake ≥175 grams (equivalent to 5,250 milliliters of beer/1,750 milliliters of wine), and for female adults, weekly pure alcohol intake ≥105 grams (equivalent to 3,150 milliliters of beer/1,050 milliliters of wine).

ESR (erythrocyte sedimentation rate, mm/h): Normal range: 0–15 (male), 0–20 (female) [Weiss method, standard clinical reference].

Bold fonts indicate that $P$-values are statistically significant.

a

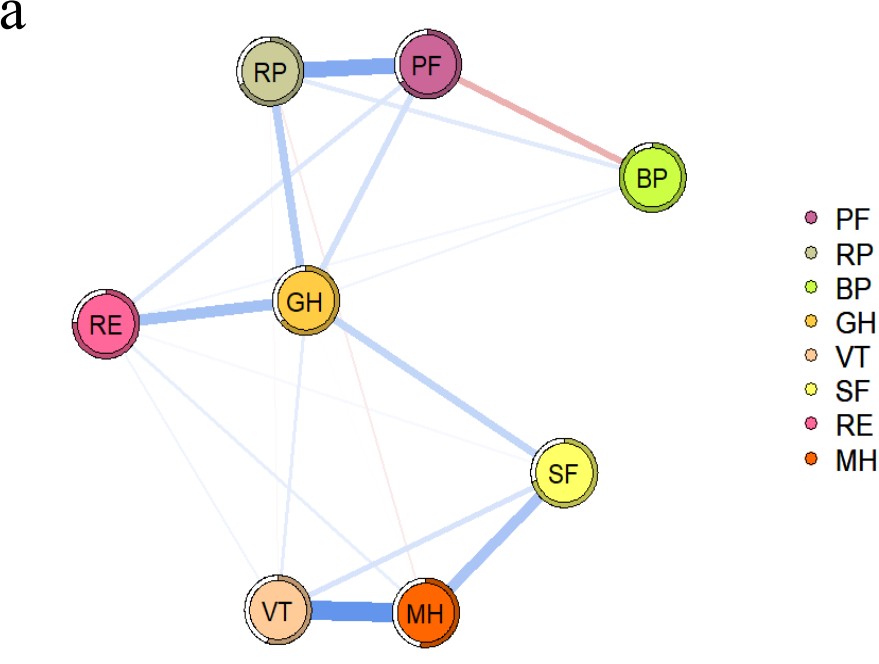

b

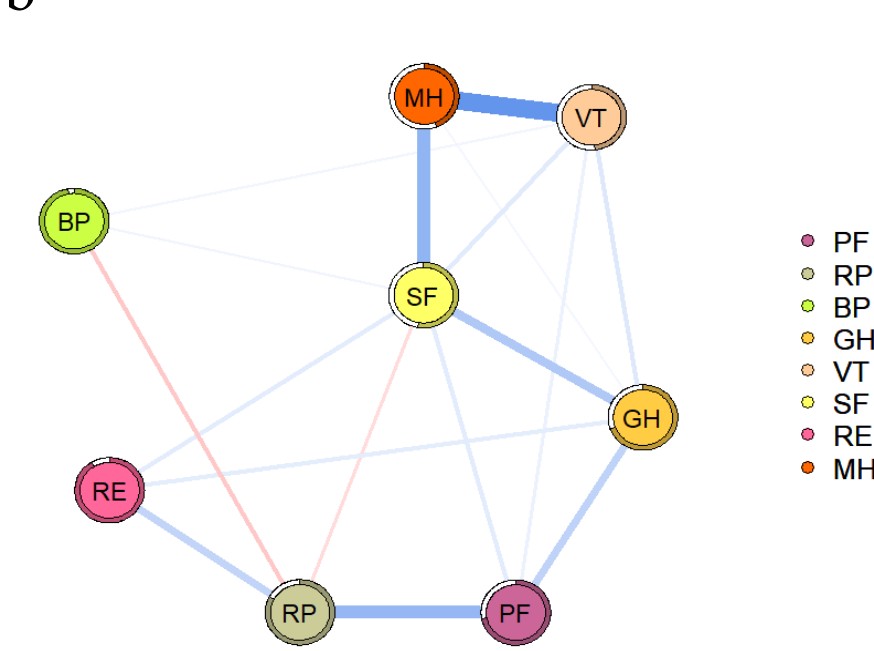

**Figure 1  QoL network diagram in RA patients without (A) and with depressive symptoms (B).** The blue line indicates a positive correlation, the red line indicates a negative correlation, and the thickness of the line reflects the correlation. PF, physical function; RP, role-physical; BP, bodily pain; GH, general health; VT, vitality; SF, social function; RE, role-emotional; MH, mental health.

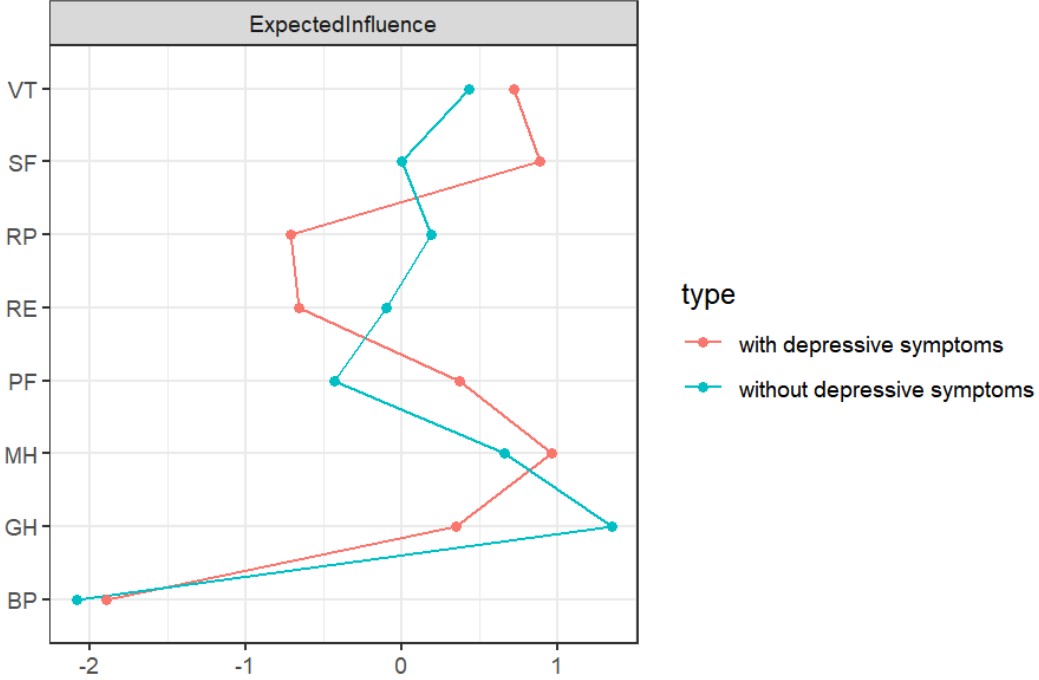

**Figure 2** **EI diagram of QoL in RA patients.** PF, physical function; RP, role-physical; BP, bodily pain; GH, general health; VT, vitality; SF, social function; RE, role-emotional; MH: mental health.

In both networks, PF and RP strongly associate with each other, as shown in Table S1. The NCT results showed no difference in the global strength of the two networks ($S = 0.201$, $P = 0.673$), but there were differences in their structure ($M = 0.274$, $P = 0.029$) and some specific edge strengths. Further analysis showed that the edges MH-PF, MH-RP, MH-BP, MH-GH, MH-VT, RP-VT, and GH-VT were more strongly associated in the network with depressive symptoms.

The nodes in the two networks with the highest EI values are different, as shown in Fig. 2. MH (1.100) has the highest EI value in the network with depressive symptoms, followed by SF (1.000) and VT (0.960). GH (1.200) has the highest EI value in the network without depressive symptoms, followed by MH (0.950) and VT (0.870). Figure S1 provides specific values.

Both network structures showed ideal accuracy and stability. The CS-C of the EI was 0.749 in the network without depressive symptoms and 0.673 in the network with depressive symptoms (see Fig. S2). The results of nonparametric bootstrapping showed that the 95% CIs of the edge weights were narrow and highly accurate (see Fig. S3).

As shown in Table 2, RA patients with depressive symptoms had lower scores on all QoL domains than those without depressive symptoms. In both networks, MH and VT have the highest predictability, meaning that the surrounding nodes can explain more than half of the variation in these two nodes.

**Table 2  Means plus or minus standard deviation and predictability of domains of QoL.**

| Domains | Without depressive symptoms ($n = 235$) | | With depressive symptoms ($n = 189$) | |
|---------|---------------|----------------|---------------|----------------|
| | M ± SD | Predictability | M ± SD | Predictability |
| PF | 64.57 ± 28.12 | 0.607 | 35.76 ± 29.61 | 0.502 |
| RP | 47.55 ± 42.76 | 0.557 | 10.44 ± 27.16 | 0.338 |
| BP | 76.12 ± 8.97 | 0.507 | 73.44 ± 8.65 | 0.053 |
| GH | 53.76 ± 15.51 | 0.702 | 31.82 ± 14.41 | 0.529 |
| VT | 68.72 ± 13.18 | 0.824 | 47.51 ± 17.00 | 0.773 |
| SF | 66.93 ± 15.74 | 0.713 | 43.49 ± 24.19 | 0.708 |
| RE | 68.22 ± 40.67 | 0.514 | 11.81 ± 28.89 | 0.147 |
| MH | 74.11 ± 13.71 | 0.847 | 48.48 ± 19.85 | 0.808 |

Notes.

PF, physical function; RP, role-physical; BP, bodily pain; GH, general health; VT, vitality; SF, social function; RE, role-emotional; MH, mental health.

Figure 3 shows the relationship between disease activity and QoL domains in RA patients. Disease activity is directly related to SF, VT, GH, BP, RP, and PF, with PF (edge weight = 0.237) having the strongest association with disease activity.

## DISCUSSION

First, we performed a network comparison of QoL between RA patients with and without depressive symptoms and found differences in network structure and some specific edge strengths. In exploring the relationship between disease activity and QoL domains, PF was found to have the strongest association with disease activity.

Our study found that MH, SF, and VT showed stronger associations in the network with depressive symptoms, and MH played an important role. *Lempp et al. (2011)* showed that both RA and depression can be detrimental to a patient's mental health, but depression declines more rapidly in the mental health domain. When a patient's mental health is impaired, it may affect their cognitive abilities and prevent them from interacting properly with others (*Dobrek & Głowacka, 2023; Iaquinta & McCrone, 2015*). According to the Attachment Model Theory, the inability to develop strong and lasting emotional connections with others can be one of the reasons for depressive symptoms (*Bernaras, Jaureguizar, & Garaigordobil, 2019*). As the depressive symptoms persist, the patient's interest in things diminishes, which may affect their vitality (*Lempp et al., 2011*).

PF and RP showed strong associations in both networks. As the disease progresses, RA can lead to joint damage and even disability, which has a significant impact on a person's health, especially in the domain of physical function (*Lempp et al., 2011*). Limitations in physical function can be associated with increased difficulty in completing daily activities and work (*Citera et al., 2015*). For example, some manual workers may not be able to perform heavy physical activities as normally. Even cerebral workers may be unable to perform fine hand activities, such as keyboarding. This may explain a strong association between PF and RP in patients with RA.

Significant negative association between disease activity and QoL, consistent with previous studies (*Gouda et al., 2023; Qorolli et al., 2019*).In addition, we found that PF

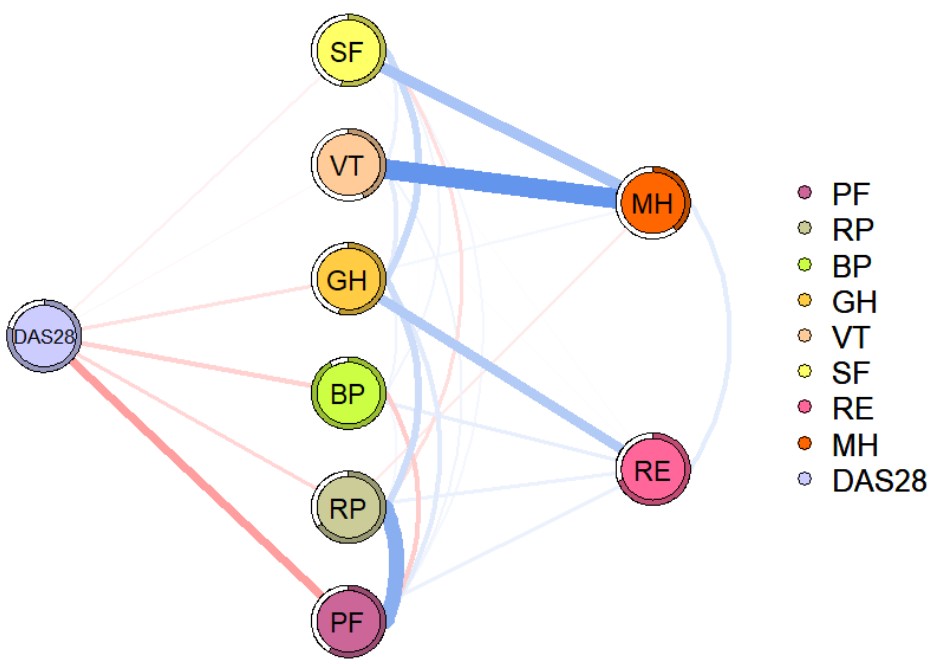

**Figure 3 Flowchart of disease activity and QoL in patients with RA.** The blue line indicates a positive correlation, the red line indicates a negative correlation, and the thickness of the line reflects the correlation. DAS28, disease activity; PF, physical function; RP, role-physical; BP, bodily pain; GH, general health; VT, vitality; SF, social function; RE, role-emotional; MH, mental health.

was most strongly associated with disease activity. Disease activity is often accompanied by elevated levels of inflammation, which may produce a range of clinical symptoms, including joint pain, joint swelling, morning stiffness, and even systemic symptoms that limit physical function (*Feng et al., 2024*; *Parrey, Koka & Ismail, 2024*). It's worth noting that patients with RA, influenced by limitations in physical function, may reduce activity, leading to a decrease in skeletal muscle content and an increase in fat content. Fat content has been associated with an increased inflammatory load, which can lead to increased levels of inflammation and disease activity in turn (*Khoja et al., 2018*; *Sokka et al., 2008*).

To our knowledge, this is the first study to explore the relationship between disease activity and QoL in RA patients using a network analysis approach. However, this study has some limitations. First, we recruited 424 RA patients from the First Affiliated Hospital of Anhui Medical University in Hefei, China; their small sample size does not allow for widespread generalization. Second, the cross-sectional study design didn't permit causal inferences. This study can only provide potential hypotheses for future intervention studies and longitudinal studies. Third, we assessed QoL and depressive symptoms using self-rating scales; future studies may consider more objective assessment methods. Furthermore, the lack of the HADS anxiety subscale in this study limited its ability to assess comorbidity, which may have led to an underestimation of the psychological burden. Fourth, we did not collect treatment-related variables; treatments (*e.g.*, traditional synthetic disease-modifying antirheumatic drugs/csDMARDs, biologics/bDMARDs, targeted synthetic

disease-modifying antirheumatic drugs/tsDMARDs, glucocorticoids, *etc.*) may directly or indirectly affect patients' QoL and depressive symptoms. Fifth, the study design does not include a control group for comparison, which may differentiate the effects of the intervention from other confounding variables. To overcome these limitations, future studies could consider controlled trials with extended follow-up periods to strengthen causal inferences and assess long-term effects.

## CONCLUSION

Disease activity is strongly associated with QoL in RA patients, especially in the domain of PF, and the structure of the QoL network changes in the presence of depressive symptoms in RA patients.

### Funding

This work was supported by the Key Scientific Research Foundation of the Education Department of Anhui Province (No. 2023AH050601) and the Natural Science Foundation project of Anhui Province (No. 2408085MH216). The funders had no role in study design, data collection and analysis, decision to publish, or preparation of the manuscript.

### Grant Disclosures

The following grant information was disclosed by the authors:
Key Scientific Research Foundation of the Education Department of Anhui Province: No. 2023AH050601.
Natural Science Foundation project of Anhui Province: No. 2408085MH216.

### Competing Interests

The authors declare there are no competing interests.

### Author Contributions

- Ruo-Wei Ma conceived and designed the experiments, performed the experiments, analyzed the data, authored or reviewed drafts of the article, and approved the final draft.
- Lin Zhu conceived and designed the experiments, authored or reviewed drafts of the article, and approved the final draft.
- Ming-Hui Zhang conceived and designed the experiments, authored or reviewed drafts of the article, and approved the final draft.
- Cui-E Li performed the experiments, analyzed the data, prepared figures and/or tables, authored or reviewed drafts of the article, and approved the final draft.
- Yu-Xuan Zhang performed the experiments, analyzed the data, prepared figures and/or tables, authored or reviewed drafts of the article, and approved the final draft.
- Bin-Bin Feng performed the experiments, analyzed the data, prepared figures and/or tables, authored or reviewed drafts of the article, and approved the final draft.

- Guo-Cui Wu analyzed the data, authored or reviewed drafts of the article, and approved the final draft.

## Human Ethics

The following information was supplied relating to ethical approvals (i.e., approving body and any reference numbers):

All participants in this study gave informed consent, and all subjects provided written informed consent. The Anhui Medical University Ethics Committee approved this study under ethical number 83243410.

## Data Availability

Raw data is available in the Supplemental Files.

## Supplemental Information

Supplemental information for this article can be found online at http://dx.doi.org/10.7717/peerj.19907#supplemental-information.

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
