# Peer review of "The relationship between disease activity and quality of life in rheumatoid arthritis patients: a network analysis"

_PeerJ, doi:10.7717/peerj.19907_

## Round 0.1 · original submission · Major Revisions

· Academic Editor

Major Revisions

·

Basic reporting

please see Additional comments.

Experimental design

please see Additional comments.

Validity of the findings

please see Additional comments.

Additional comments

1. Nice work, refreshing to read.
2. “The EULAR guidelines set low disease activity as the treatment goal for RA patients to alleviate disease progression” – first, please note that EULAR sets remission as the target of RA treatment and, when remission is not feasible, low disease activity is accepted; second, disease progression is not “alleviated”, it is “stopped”, “inhibited”, “reduced”.
3. “We conducted a cross-sectional study involving 424 patients with RA recruited from the First Affiliated Hospital of Anhui Medical University in Hefei, China.” – this information from the methods reports the number of included patients. The number of included patients is a result of the methods, so please report it in the Results section. Unless, you decided at the beginning of the study to include exactly 424 RA patients for reasons unknown. Please also see line 129.
4. “We conducted a cross-sectional study involving 424 patients with RA recruited from the First Affiliated Hospital of Anhui Medical University in Hefei, China.” – recruited how? Did you screen all the RA patients coming to the First Affiliated Hospital of Anhui Medical University in Hefei within the reported time period? If not, how did you choose who to ask to join? This is important for randomness of inclusion.
5. “a diagnosis based on the 2010 American College of Rheumatology (ACR)/European League Against Rheumatism (EULAR) classification criteria and scoring system for RA” – First, please be aware that these criteria are for classification, not for diagnosing patients: a patient fulfilling these criteria may not have RA, and conversely, a patient diagnosed with RA may not fulfill these criteria. RA is diagnosed based on medical history, clinical interview, clinical examination, imaging and laboratory tests. So please rephrase to say that your patients diagnosed with RA also fulfilled those criteria. Second, even though the criteria use scoring, they are not a “scoring system for RA”, they are simply “RA classification criteria”.
6. “joint VAS score, number of swollen joints, number of joint pressures, and ESR” – what does “VAS” and “ESR” mean? Please recheck abbreviation management in the entire document: abbreviate a term the first time you use it, then use only the abbreviation.
7. General information included age, gender, body mass index (BMI), family location, marital status, education level, smoking, drinking, disease duration, number of chronic diseases, sleep duration, joint VAS score, number of swollen joints, number of joint pressures, and ESR” – there were no treatment variabiles collected, why? This should be discussed in the limitation of the study, since treatments (csDMARDs, bDMARDs, tsDMARDs, glucocorticoids) can directly and indirectly influence quality of life and depression.
8. “Refer to the DAS28 scale proposed by Prevoo[24]. DAS28 = 0.56 × sqrt (number of joints with pressure pain) + 0.28 × sqrt (number of swollen joints) + 0.7 × ln(ESR) + 0.014 × PGA (PGA: VAS 0-100mm). DAS28 Score: DAS28 < 2.6 represents that the disease is in remission; 2.6 DAS28 3.2 represents low disease activity; 3.2 < DAS28 5.1 represents moderate disease activity; DAS28 > 5.1 represents severe disease activity.” – this text looks like notes taken by students at a course. Please write proper academic text. This means using sentences and phrases, with a subject, predicate etc.
9. According to the methods section, your patients underwent clinical interview, clinical examination, blood sampling and questionnaire filling. Were all of these procedures done on the same day for each patient?
10. “Descriptive analyses of the variables were performed using SPSS 26.0.” - please note that the software is not properly cited. Please see: https://www.ibm.com/support/pages/how-cite-ibm-spss-statistics-or-earlier-versions-spss
11. “The participants were 58.21 ± 12.41 years old” – Is this a mean with a standard deviation? Please include in the Statistics section a statement on how you report your data and why.
12. Table 1 – please indicate what is the measurement scale of age (years), BMI (kg/m2).
13. Table 1 – please replace “countryside” with “rural” and “city” with “urban” for the sake of academic style.
14. Table 1 – First, please replace “drinking” with “alcohol consumption”. Second, define the term: what does “drinking yes” mean? A bottle of wine a day? A glass of scotch on Sundays?
15. Table 1 – “Number of joint pressures” – this does not make sense in English. Do you mean to say tender joints? Please rephrase.
16. Table 1 – please add normal ranges for ESR.
17. Table 2 – the number under “With depressive symptoms” – “M±SD” is unintelligible.
18. Reference 14 is broken.
19. Reference 24 has unknow text at the end.

·

Basic reporting

no comments

Experimental design

- Did the study exclude patients having fibromyalgia or other systemic diseases affecting sleep quality? If so, you should fully mention that in the exclusion criteria.
- How were the patients selected (e.g., consecutively, randomly, or selectively)?
- More details should be provided in terms of the IRB approval.

Validity of the findings

- Failure to use a control group can make it impossible to draw meaningful conclusions from a study.
- Disease Activity Score 28 joints (DAS28): Who assessed this disease activity tool? Is it a rheumatologist? Is it a blinded investigator? Which DAS28 was used, ESR or CRP? Please elaborate.

Additional comments

- Kindly include the following limitations: lacks of Control group, Furthermore, there is a lack of subsequent follow-up time

Reviewer 3 ·

Basic reporting

No comment

Experimental design

There are some things i would like to comment:
1. HADS is a scale to measure anxiety and depression. Using HADS, a person can have both anxiety and depression. The authors only used the depression subscale. This will lead to some bias and confounding factors, because in reality the subjects could have depression only (anxiety subscale is normal) or depression mixed with anxiety (both subscales showed abnomal score). Would it not be better if the authors used both subscale, then analyze separately the subjects with only depression symptoms and subjects with mixed anxiety-depression symtoms?
2. The authors decided to use the cut off score of greater than 7 (mild). I think that the cut off point is very low and would overstate the depression symptoms in RA patient. The cut off point of greater than 11 seems reasonable for me. Please elaborate why the authors decided to use a very low cut off point.

Validity of the findings

No comments

Additional comments

This is a very interesting study. I would like to point out that it would be more informative if the authors explored deeper the relationship between disease activity and QOL. Sometimes the patient already had remission or low disease activity, but they would still have depression symptoms. This study might benefit by exploring this subgroup of patient and which domain in QOL that is the most significantly related to the depression symptoms?

---

## Round 0.2 · accepted · Accept

· Academic Editor

Accept

I have carefully reviewed your manuscript along with the comments provided by the reviewers.

·

Basic reporting

The authors have adequately addressed all the raised issues.

Experimental design

-

Validity of the findings

-